# Multiscale modeling of blood circulation with cerebral autoregulation and network pathway analysis for hemodynamic redistribution in the vascular network with anatomical variations and stenosis conditions

Jiawei Liu[1]*, Atsushi Kanoke[2,3], Hidenori Endo[3], Kuniyasu Niizuma[4,5], Hiroshi Suito[1,6]

**1** Advanced Institute for Materials Research, Tohoku University, Sendai, Japan, **2** Department of Neurosurgery, Kohnan Hospital, Sendai, Japan, **3** Department of Neurosurgery, Tohoku University Graduate School of Medicine, Sendai, Japan, **4** Department of Translational Neuroscience, Tohoku University Graduate School of Medicine, Sendai, Japan, **5** Department of Neurosurgical Engineering, Graduate School of Biomedical Engineering, Tohoku University, Sendai, Japan, **6** Mathematical Science Center for Co-creative Society, Tohoku University, Sendai, Japan

\* liu.jiawei.e2@tohoku.ac.jp

## Abstract

Cerebral hemodynamics is fundamentally regulated through the Circle of Willis (CoW), which redistribute flow via communicating arteries to stabilize perfusion under anatomical variations and vascular stenosis. In this study, a multiscale circulation model was developed by coupling a multiscale systemic hemodynamic framework with cerebral arterial network reconstructed from medical imaging. The model integrates a cerebral autoregulation mechanism (CAM) and enables quantitative simulation of flow redistribution across the CoW under normal, anatomically varied, and pathologically narrowing (stenosis) conditions. Baseline simulations at normal states reproduced physiological flow distributions in which the communicating arteries remained nearly inactive, showing negligible across-flow and agreement with clinical measurements, while two anatomical variations revealed distinct collateral activation patterns: the anterior communicating artery (ACoA) acted as the earliest and most sensitive functional collateral pathway, whereas the posterior communicating arteries (PCoAs) exhibited structure-dependent engagement. Progressive stenosis modeling further demonstrated the transition from a complete CoW to a fetal-type posterior cerebral artery (PCA) configuration, with early ACoA flow reversal followed by the ipsilateral PCoA activation, in agreement with experimental and transcranial Doppler observations. We further present a path-based quantitative analysis of source-to-sink flow contributions to quantitatively illustrates how the cerebral vascular network dynamically reconfigures collateral pathways in response to structural changes. Overall, the proposed framework provides a physiologically interpretable and image-informed tool for investigating cerebral flow regulations through the functional

**Data availability statement:** Data Availability: The MR imaging data and the associated cerebral vascular network files used in this study are publicly available from the Zenodo repository (https://zenodo.org/record/3707179), as reported by Ii et al. (2020). The source code and data files underlying the findings reported in this study are publicly available at a Github database link (https://github.com/jw-liu/cerebral-autoregulation-model). These URLs provide direct access to the data used in this study, and appropriate citation guidelines should be followed.

**Funding:** This work was supported by the Japan Science and Technology Agency (JST) Moonshot R&D Program (Grant Number JPMJMS2023 to JL, KN, and HS). The funders had no role in study design, data collection and analysis, decision to publish, or preparation of the manuscript.

**Competing interests:** The authors have declared that no competing interests exist.

collaterals within the CoW, offering potential for clinical applications in the diagnosis, prognosis, and treatment planning of cerebrovascular diseases.

## Author summary

Cerebral blood circulation is regulated by complex interactions between vascular structure, flow dynamics, and autoregulatory mechanisms. Understanding how blood flow redistributes under anatomical variations and pathological conditions, such as arterial stenosis, remains a major challenge. In this study, we develop a multiscale computational model of the human cardio-cerebrovascular system that integrates a lumped-parameter heart model, a cerebral arterial network, and a cerebral autoregulation mechanism. The model allows us to simulate hemodynamic responses under both normal and pathological conditions and to investigate how collateral pathways within the Circle of Willis contribute to maintaining cerebral perfusion. In addition to simulating blood flow, we introduce a pathway-based analysis that decomposes flow into source-to-sink contributions, enabling quantitative identification of how different inflow arteries supply specific brain regions. This approach provides new insights into the mechanisms of flow redistribution and collateral circulation. Our results demonstrate that cerebral autoregulation plays a critical role in stabilizing perfusion and shaping flow pathways under anatomical variations and stenosis conditions. The proposed framework offers a useful tool for studying cerebral hemodynamics and may help support the assessment of surgical strategies and treatment planning in cerebrovascular diseases.

## 1. Introduction

For the high metabolic demand of the brain and its sensitivity to ischemia, cerebral blood flow (CBF) is believed be tightly regulated to ensure adequate oxygen and nutrient supply to neural tissue [1,2]. The Circle of Willis (CoW), an arterial ring connecting the anterior and posterior cerebral circulations, plays a central role in maintaining cerebral perfusion stability. It provides collateral pathways through the anterior communicating artery (ACoA) and posterior communicating arteries (PCoAs), enabling redistribution of blood flow when regional perfusion is impaired due to vascular narrowing (occlusion or stenosis). Specifically, interhemispheric blood flow across the ACoA, often accompanied by flow reversal in the proximal anterior cerebral artery, provides an effective cross-hemispheric collateral pathway, while PCoAs mediate bidirectional compensation between the anterior and posterior circulations depending on local hemodynamic conditions [3,1].

However, it is demonstrated that the compensatory capacity of the CoW is highly dependent on its anatomical completeness [4,5]. Lippert and Pabst [6] have reported that only about half of the population has a morphologically complete CoW, while the remaining exhibit hypoplasia or absence of one or more arteries, characterized

by very small diameters or incomplete development. These anatomical variations can significantly alter cerebral hemodynamics, reduce collateral availability, and therefore increase the risk of ischemic events when major arteries, such as the internal carotid artery, become narrowed or occluded [7,8]. Experimental studies [9] have further clarified the collateral mechanism of the CoW. Therefore, understanding how the CoW within the realistic vascular network dynamically redistributes blood flow under different anatomical variations and pathological conditions is crucial for both physiological interpretation and surgery planning.

In recent decades, computational modeling has emerged as a powerful approach to investigate cardiovascular and cerebrovascular diseases, and to develop non-invasive diagnosis [10,11]. One-dimensional (1D) hemodynamic models have been widely adopted for simulating pressure and flow wave propagation in compliant arterial networks [12,4]. These models provide an efficient and physiologically consistent description of hemodynamics in the large blood vessels, while allowing the investigation of how geometric and resistive changes affect perfusion patterns throughout the cerebral vascular network. Alastruey et al. [4] modeled blood flow in the CoW, demonstrating how CoW configurations and occlusions affect the cerebral flow and identifying the ACoA being a more essentially collateral pathway than the PCoAs when the ICA is occluded. More recently, [5,13] developed a multiscale model coupled with a machine-learning-based pressure-drop model and a cerebral autoregulation mechanism (CAM), predicting hyperperfusion risks following bypass surgery in patients with severe carotid stenosis.

Despite these advances, most existing CoW models focus on idealized geometries or steady post-occlusion (or severe stenosis) conditions, neglecting the progressive nature of arterial narrowing and the gradual redistribution of flow across the cerebral network. In reality, cerebral autoregulation acts as a global mechanism, continuously adjusting vascular resistance in response to systemic and regional pressure changes. Consequently, accurately characterizing how the CoW and its downstream vessels cooperatively adapt to topological and resistive changes remains a major challenge.

To overcome these limitations, this study develops a 0D–1D multiscale circulation model that coupled systemic and cerebral hemodynamics with a CAM on a realistic cerebral arterial network reconstructed from the medical images. The proposed framework enables dynamic simulation of flow redistribution across the entire cerebral vasculature under normal, anatomical variation, and progressive stenosis conditions. This model quantitatively evaluates collateral responses through the ACoA and PCoAs, providing insights into how cerebral blood flow dynamically adapts to variations of cerebral network.

This paper is organized as follows. In Section 2, we introduce the governing equations and the formulation of the 0D–1D multiscale circulation model, incorporations with CAM and stenosis model, and the sparse inversions of path-flow. Section 3 presents the implementation details and computational results for the simulation configurations under the baseline, anatomical variations, and progressive stenosis conditions, with subsequent discussions in Section 4. Finally, Section 5 presents conclusions and future prospects.

## 2. Mathematical models of the blood circulation network

In this study, we implemented a coupled 0D-1D framework to provide a physiologically consistent representation of cardio-cerebrovascular circulation reconstructed from medical images (Fig 1). Within this framework, cerebral autoregulation mechanism (CAM) is incorporated to dynamically and globally adjust vascular resistance, along with pathological conditions (anatomical variations or stenosis) are modeled by structural modifications of the vascular network or by introducing localized pressure drops, enabling quantitative analysis of flow redistribution. For clarity, a complete list of symbols and abbreviations used in the model is provided in the nomenclature table in Supporting Information (S1 Table).

### 2.1. Modeling of 0D-1D cardio–cerebrovascular networks with CAM

In the multiscale framework, large arteries in the extracted cerebral vascular network are modeled as axisymmetric and deformable tubes, through which blood flow is governed by the 1D Navier-Stokes equations. When passing through a

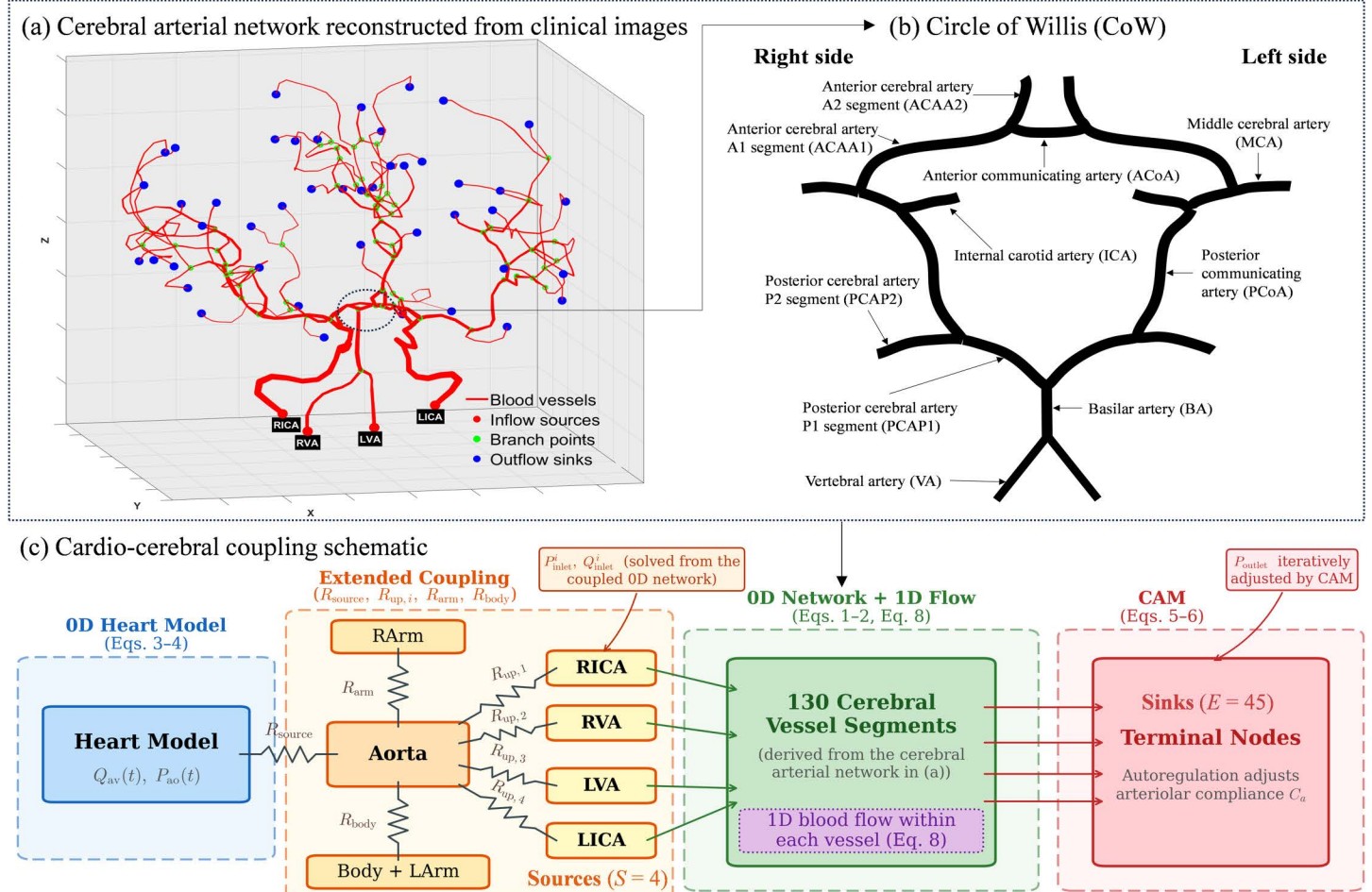

**Fig 1. Schematic representation of the multiscale model.** (a) Cerebral arterial network reconstructed from clinical images. Centerlines are extracted and circular cross-sections are used as estimates for the vessel radius. The 1D arterial tree is coupled to the heart model at inflow sources (red points), branch (green) points are treated using Riemann invariants, and outflow sinks (blue points) are linked to 0D terminal models. The segment thickness are normalized to its radius to improve visibility. (b) Detailed vascular anatomy of the CoW, including the anterior, middle, and posterior cerebral arteries and their communicating branches. (c) A resistance-based coupling scheme between the heart model and the cerebral arterial network. The 0D heart model (Eqs. 3–4) derives the aortic flow $Q_{av}(t)$ and pressure $P_{ao}(t)$. These outputs are distributed through an extended resistance network consisting of $R_{arm}$, $R_{body}$, and $R_{up,i}$, which connects the aortic arch to the right arm (RArm), the systemic circulation (body + left arm), and the cerebral arterial network (a) via four inlet segments: right internal carotid artery (RICA), right vertebral artery (RVA), left vertebral artery (LVA), and left internal carotid artery (LICA). In this framework, these inlet arteries act as inflow sources of the arterial network (marked by red filled circles in (a)), while the distal terminal nodes correspond to outflow sinks (marked by blue filled circles in (a)) whose outlet pressures $P_{outlet}$ are regulated by CAM through iteratively adjusting distal vascular resistance and compliance. The iterative coupling procedure is described in Section 2.1 and illustrated in Fig 2.

vascular junction, flows across it are coupled via characteristic (Riemann-invariant) relations, whereas the inflow and out-flow segments within the network are respectively connected to the 0D heart model and the peripheral circulations represented by 0D lumped-parameter models.

The modeled cerebral arterial network (as depicted in Fig 1a), along with the anatomical and geometric parameters used to define its structure and function, was extracted from vessel segmentations of computed tomography (CT) images [14]. The reconstructed cerebral network contains the CoW, which is a ring of interconnected arteries at the base of the brain and shown in Fig 1b. It connects the anterior and posterior circulations and the left and right hemispheres, providing

immediate diversion of blood flow in the acute occlusion cases. Fig 1c summarize the coupling of the whole-body and heart-cerebral circulations.

We introduced the CAM to analyze the blood flow direction changes and redistribution under different pathological scenarios, providing patient-specific adaptability. The following introduces the detailed mathematical formulation of this multiscale model.

**2.1.1. 0D vascular resistance network for cerebral arteries.** Cerebral autoregulation ensures stable cerebral blood flow despite fluctuations in systemic blood pressure. The mathematical model employed in this study is based on a network representation of the arterial system, in which each vessel segment is modeled as a resistance, and the flow rate $Q$ across the segment is governed by the pressure drop $\Delta P$ according to Poiseuille's law [15]:

$$P_i - P_j = \Delta P_{ij} = Q_{ij}R_{ij}, \quad R_{ij} = \frac{8\mu l_{ij}}{\pi r_{ij}^4}, \quad i \in \{1, \dots, N_{node}\}, \; j \in A(i).$$

(1)

Here, $\mu$ is the blood viscosity. Let $N_{node}$ denote the total number of nodes in the vascular network, and $A(i)$ is the set of nodes directly connected to node $i$. For any pair $(i, j)$ with $j \in A(i)$, $l_{ij}$, $r_{ij}$, and $R_{ij}$ are respectively the length, radius, and vascular resistance of the segment connecting the two nodes $i$ and $j$.

Next, we aim to achieve overall pressure equilibrium using iterative computations. The cerebral arterial network (Fig 1a) is represented by a nodal resistance network with multiple inlet sources and terminal outlets. With the flow conductance $G_{ij} = 1/R_{ij}$ defined for directly connected node pair $(i, j)$ with $j \in A(i)$, the pressure distribution across the vascular network is determined by mass conservation at any node $k$:

$$\sum_{j \in A(k)} G_{kj} \left( P_k - P_j \right) = Q_k,$$

where the net nodal flow $Q_k$ is obtained by summing the flows $Q_{kj}$ through all vessel segments connected to node $k$ as defined in Eq. (1). By separating these connecting nodes into interior and boundary nodes, we obtain

$$\left( \sum_{j \in A(k)} G_{kj} \right) P_k - \sum_{j \in A_{int}(k)} G_{kj}P_j = \sum_{j \in A_b(k)} G_{kj}P_j + Q_k$$

with

$$A(k) = A_{int}(k) \oplus A_b(k),$$

where $A_{int}(k)$ and $A_b(k)$ are subsets of $A(k)$ corresponding to interior and boundary nodes, respectively. It is worth noting that due to conservation of mass, the flow term $Q_k$ should be zero for all nodes other than the inlet and outlet nodes within the network. In this way, the pressure distribution across the entire cerebral vascular network is determined by the mass-balance equation system:

$$BP = G_bP_b + Q_b,$$

(2)

where the conductance matrix $B$ is defined as

$$B_{mn} = \begin{cases} \sum_{j \in A(m)} G_{mj}, & m = n \\ -G_{mn}, & m \neq n, n \in A_{int}(m) \\ 0, & \text{otherwise} \end{cases}$$

and the conductance matrix $\mathbf{G}_b$ associated with boundary nodes is defined as

$$(G_b)_{mn} = \begin{cases} G_{mn}, & n \in \mathbf{A}_b(m), \\ 0, & \text{otherwise.} \end{cases}$$

Here, the pressure and flow rate vectors $(P_b, Q_b)$ correspond to the boundary nodes, including $S$ inlets and $E$ outlets as shown in Fig 1c. The inlet boundary conditions are determined by the upstream resistance network that distributes the cardiac output modeled by the 0D heart model, while the outlet pressure at terminal nodes are regulated by the CAMs, which are introduced in the following subsections.

**2.1.2. Coupling the 0D vascular network with the 0D heart model.** The systemic inflow conditions of the cerebral network are provided by a lumped 0D heart model, which drives the entire closed-loop system as illustrated in Fig 1c. Several 0D models for simulating this circulation have been reported in detail by Sun et al. [16] and Liang et al. [17]. In this study, the time-varying elastance model and valve dynamics are employed for providing physiologically realistic pulsatile pressure and flow waveforms as inflow into the cerebral vascular network while maintaining low computational cost. The model simulates $Q_{av}$, which represents the flow rate from the left ventricle to the aorta, as follows:

$$\frac{dQ_{av}}{dt} = \frac{P_{lv} - P_{ao} - R_{av}Q_{av} - B_{av}Q_{av}|Q_{av}|}{L_{av}} \times D_{av}(P_{lv}, P_{ao})$$

$$\frac{dV_{ra}}{dt} = Q_{vc} - Q_{tv},$$

(3)

with

$$D_{av}(P_{lv}, P_{ao}) = \begin{cases} 1 & P_{lv} > P_{ao}, \\ 0 & P_{lv} \leq P_{ao}. \end{cases}$$

Here, $Q_{tv}$ and $Q_{vc}$ denote the flow rates through the tricuspid valve and the vena cava, respectively. $R_{av}$ represents the viscous resistance of the aortic valve, and $B_{av}$ is the Bernoulli resistance of the valves, which is associated with convective acceleration and dynamic pressure losses. $L_{av}$ and $R_{av}$ respectively signify the coefficients for inertial terms and viscous losses. Cardiac valve function $D_{av}$ is defined as an ideal diode model, which opens and closes instantaneously depending on the changing of the pressure gradient sign between the left ventricle ($P_{lv}$) and aorta ($P_{ao}$). The blood pressure ($P_{cm}$) in the four cardiac chambers including the right atrium ($ra$), right ventricle ($rv$), left atrium ($la$), and left ventricle ($lv$), is obtained by [17,18]

$$P_{cm}(t) = (E_{cm,A}e(t) + E_{cm,B})(V_{cm} - V_{cm,0}) + S_v \frac{dV_{cm}}{dt}, \quad cm = \{ra, rv, la, lv\},$$

(4)

where $E_{cm,A}$ and $E_{cm,B}$ respectively represent the active and passive elastances of the four cardiac chambers. In addition, $V_{cm}$ stands for the current chamber volume. $V_{cm,0}$ denotes the dead volume, assumed here as 0. $S_v$ expresses the visco-elasticity coefficient of cardiac wall and $e(t)$ signifies the normalized time-varying elastance. The detailed formulations and parameter values are shown in Supporting Information (S2 Text and S1 Table).

**2.1.3. Coupling the 0D vascular network with a CAM.** While the 0D heart model provides the inflow boundary conditions for the 0D vascular resistance network, incorporating autoregulatory response at the outlets is important to physiologically simulate cerebral perfusion stabilized. Therefore, we wholly adopt the iterative CAM algorithm proposed by

[15] as the outflow boundary condition for each outlet segment of our constructed 0D vascular network. In their framework, the cerebral vascular bed at each terminal node is modeled as a passive resistance, and provided an iterative scheme to relate the distal pressure $P_{outlet}$ at iteration step $k$ to the arteriolar compliance $C_a$ and flow rate $Q_{outlet}$ at iteration step $k$-1 by:

$$\left[ \frac{P_{outlet,k} - P_v - Q_{outlet,k-1}R_v}{Q_{outlet,k-1}} \right] \cdot \left[ C_{a,k}\left(P_{outlet,k} - 2P_{ic} + P_v + Q_{outlet,k-1}R_v\right) - C_{a,0}\left(2P_{outlet,k} - 2P_{ic} - Q_{outlet,0}R_{sa}\right) \right]^2$$
$$+ 2V_{sa}^2 - 4R_{sa}V_{sa}^2 = 0 \tag{5}$$

with

$$C_{a,k} = C_{a,0} + \frac{1}{2} \begin{cases} \Delta C_a^+ tanh\left[ \frac{2G_q}{\Delta C_a^+}\left(1 - \frac{Q_{outlet,k}}{Q_{outlet,0}}\right) \right], & Q_{outlet,k} < Q_{outlet,0} \\ \Delta C_a^- tanh\left[ \frac{2G_q}{\Delta C_a^-}\left(\frac{Q_{outlet,k}}{Q_{outlet,0}} - 1\right) \right], & Q_{outlet,k} > Q_{outlet,0} \end{cases}.$$

Here, $P_v$ and $P_{ic}$ respectively denote the venous and intracranial pressure. $R_v$ and $R_{sa}$ are the venous and arteriolar resistances, respectively. $V_{sa}$ is the arteriolar volume. $G_q$ is the gain of the flow feedback mechanism. $C_{a,0}$ is the baseline arteriolar compliance, $\Delta C_a^+$ and $\Delta C_a^-$ define the upper and lower limits of compliance variation. The parameter values used to define the autoregulating part for simulation are given in Supporting Information (S2 Text and S1 Table). The iterative procedure continues until the convergence criterion is satisfied:

$$max\left(|P_{outlet,k} - P_{outlet,k-1}|\right) < \varepsilon. \tag{6}$$

Then, the mass-balance system in Eq. 2 is closed by boundary conditions at the inlets through 0D heart model (Eqs.(3)–(4)) and the outlets through CAM (Eqs. (5)–(6)). After solving it for the nodal pressure $\{P_i\}$, and therefore the flow rate in each vascular segment is given by

$$Q_{ij} = G_{ij}\left(P_i - P_j\right), \tag{7}$$

with $Q_{ij}$ positive from $i$ to $j$, uniquely determining the flow direction throughout the network.

**2.1.4. Coupling the 0D vascular network with 1D flow model.** In this subsection, we extend the 0D cerebral vascular network incorporated with CAM proposed by [15] into a multiscale (0D–1D) framework. Specifically, based on the nodal pressures simulated by the 0D models (Eqs.(2)–(6)) and the corresponding flow distribution and direction using Eq.(7), we prescribe them as boundary conditions for each 1D vessel segment and perform 1D hemodynamic simulations to capture spatial and temporal characteristics of blood flow. The 1D flow simulation $Q(x,t)$ is obtainable by averaging the incompressible Navier-Stokes equations over the cross-sectional area $A(x,t)$ under the assumptions that the blood flow is axisymmetric and that it has no swirl [17]. The well-established equations of conservation of mass and momentum is represented as [18, 19]:

$$\frac{\partial A(x,t)}{\partial t} + \frac{\partial Q(x,t)}{\partial x} = 0,$$

$$\frac{\partial Q(x,t)}{\partial t} + \frac{\partial}{\partial x}\left[ \frac{Q^2(x,t)}{A(x,t)} \right] + \frac{A(x,t)}{\rho}\frac{\partial P(x,t)}{\partial x} + \frac{8\pi\mu Q(x,t)}{\rho A(x,t)} = 0,$$

$$P(x,t) = P_e + P_0 + \frac{4\sqrt{\pi}E_s h}{3A(x)}\left[\sqrt{A(x,t)} - \sqrt{A_0(x)}\right], \tag{8}$$

where $t$ represents the time and where $x$ denotes the axial coordinate along the vessel. Variables $A(x, t)$, $Q(x, t)$, and $P(x, t)$ respectively denote the cross-sectional area of the vessel, the volume flux, and the average pressure over a cross-section. Here, $\rho$ represents the blood density, $\mu$ represents the blood viscosity, $E_s$ is the Young's modulus, and $h$ is the wall thickness. $P_e$ stands for the external pressure, $A_0(x) = A(x, t_0)$ signifies a reference cross-sectional area at a given time point $t_0$, and $P_0$ denotes the reference pressure for $A = A_0$.

Based on the simulated $A(x,t)$ by the 1D solver, the resistance $R_{ij}$ of each segment considered in the 0D vascular network can be updated at time step $k$ by

$$R_{ij}^{n+1} = (1-\gamma)R_{ij}^n + \gamma R_{ij}^{eff,n}(t), \quad 0 < \gamma \le 1, \quad \text{with} \quad R_{ij}^{eff}(t) = \int_0^l \frac{8\mu\pi}{A_{ij}(x,t)^2}\,dx,$$

(9)

where $\gamma$ is a given relaxation factor to avoid numerical stiffness. The 0D linear system is re-assembled and solved to update node pressures and edge flows, reflecting the elastic response of the vessel to flow rates.

The 0D and 1D models are coupled through a sequential iterative procedure (see Fig 2 and S1 Text). In each outer cycle, the 0D system including the heart model, the cerebral resistance network, and the CAM, is first solved over one cardiac cycle $\tau$ to determine the nodal pressures $P_i(t)$ and flow rates $Q_{ij}(t)$ at each segment. These quantities are then used as boundary conditions for the 1D blood flow solver, which computes the spatially resolved flow $Q(x, t)$ and cross-sectional area $A(x, t)$ within each vessel segment. The resulting 1D solutions are subsequently used to update the effective resistances of the corresponding vessels via Eq. (9). This bidirectional coupling for a cardiac cycle $\tau$ is repeated until reaching a quasi-steady state, in which the results of two consecutive cycles ($n$-1, $n$) become nearly identical, as defined by:

$$J_n = \frac{\sum_{\omega \in \mathbf{N}} \frac{\underset{t\in[0,\ \tau]}{max}\ |Q_\omega^n - Q_\omega^{n-1}|}{\underset{t\in[0,\ \tau]}{max}\ |Q_\omega^{n-1}|} + \frac{\underset{t\in[0,\ \tau]}{max}\ |A_\omega^n - A_\omega^{n-1}|}{\underset{t\in[0,\ \tau]}{max}\ |A_\omega^{n-1}|}}{\text{num}(\mathbf{N})} < \varepsilon,$$

(10)

Here, $\mathbf{N}$ are the set of vessels and where num($\mathbf{N}$) represents the number of elements of set $\mathbf{N}$.

## 2.2. Coupling the multiscale model with Stenosis models

While the multiscale model provides a comprehensive framework for vascular hemodynamics and can be applied to various network structures, including those describing anatomical variations that are topologically equivalent to complete occlusions, it does not explicitly account for the localized flow resistance arising from pathological narrowing (stenosis). To address this limitation, we further incorporate an experiment-based stenosis model proposed by Young and Tsai [20] and reformulated equivalently by Liang et al. [19], which characterizes the additional pressure drop across a stenotic segment with highly variable cross-sectional areas. Specifically, the stenosis model is integrated into the 0D vascular network (Eq.(2)) by expressing the effective resistance for vessel segments affected by stenosis as [19,21]:

$$R_{ij}^{eff}(Q) = \frac{\Delta P}{Q} = \frac{K_v\mu}{A_0 D_0} + \frac{K_t\rho}{2A_0^2}\left(\frac{A_0}{A_s} - 1\right)^2 |Q|.$$

(11)

Here, the subscripts 0 and $S$ respectively correspond to the reference (healthy) segment and the stenotic segment. $D_0$ denotes the healthy vessel diameter, and $L_s$ is the length of the stenosis. The remaining coefficients were obtained from an extensive series of *in-vitro* steady-flow experiments conducted by Seeley and Young [22], as:

$$K_t = 1.52, \quad K_v = 32\left(0.83L_s + 1.64D_s\right)\frac{A_0^2}{D_0 A_s^2}.$$

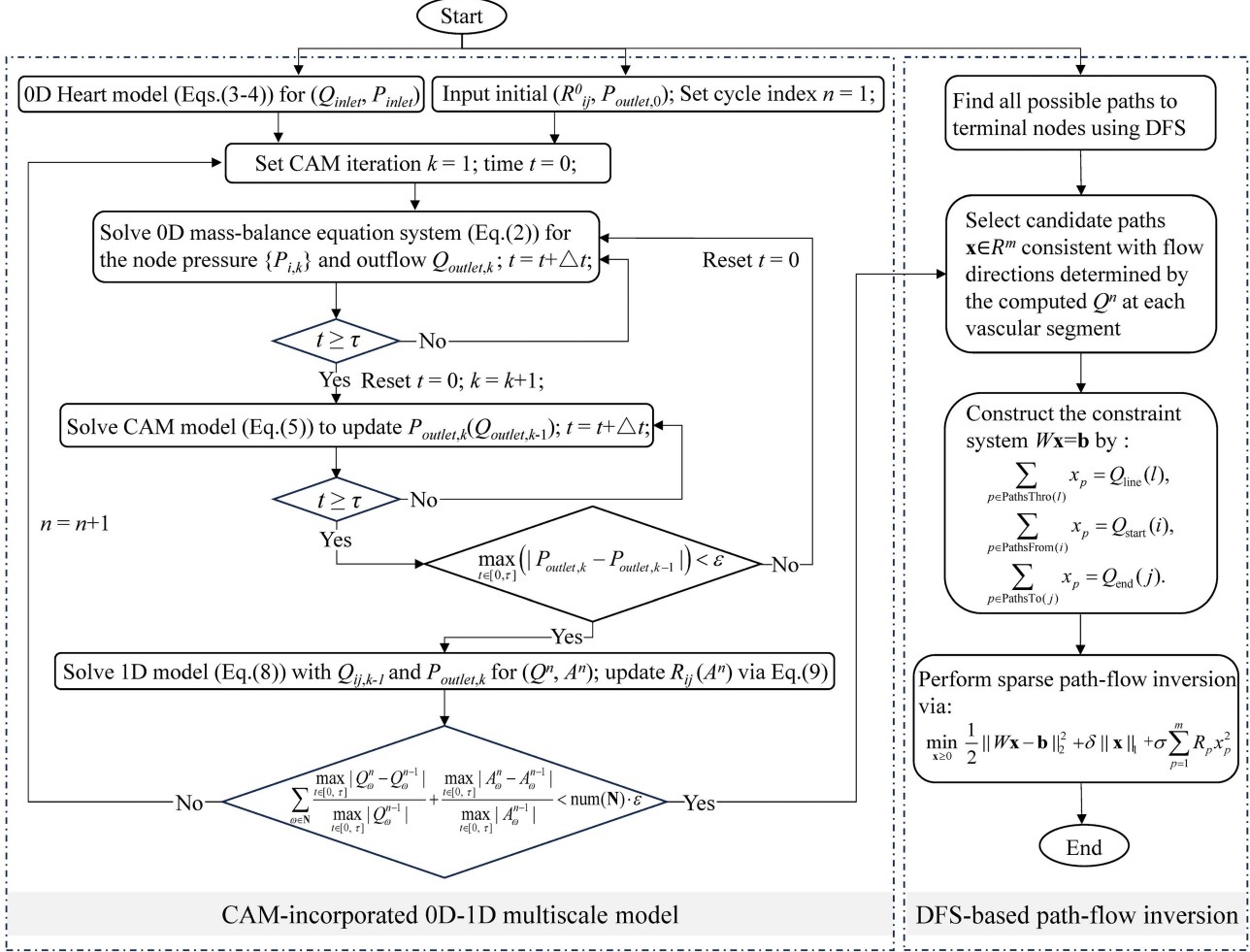

**Fig 2. Flow chart of the proposed multiscale model incorporated with CAM and the DFS-based path-flow inversion via an L1-regularized convex optimization.**

The first term in Eq.(11) accounts for viscous losses, while the second term captures inertial effects associated with geometric contraction. Moreover, the degree (severity) of stenosis $R_s$ is quantified by the percentage reduction in the vessel diameter relative to the healthy segment:

$$R_s = \left(\frac{D_0 - D_s}{D_0}\right) \times 100\%.$$

(12)

The coupling method between the two models is carried out as follows: the flow rate $Q$ and cross-sectional $A$ obtained from the 0D-1D multiscale model are used as inputs to the stenosis model. The effective resistance $R^{\text{eff}}_{ij}$, updated with the stenosis contribution, is then substituted back into the resistance coefficient matrix in Eq.(2) for subsequent computations. In the 1D solver, the stenosis is represented as a localized reduction of the reference cross-sectional area within the affected vessel segment, capturing the local hemodynamic effects of the segment narrowing. This approach enables the multiscale framework to account for the pressure losses across stenoses, thereby improving the physiological fidelity of the simulations.

## 2.3. DFS-based feasible path set and sparse inversions of path-flow

In this subsection, we introduce a post-processing analysis methodology applied to the converged hemodynamic solution, which does not alter the simulation results, but instead provides an interpretable decomposition of the simulated blood flow at the segments. It is designed to provide further insight into the computed flow distribution and to quantitatively estimate how blood flow is distributed across multiple candidate vascular paths connecting predefined source and sink nodes. This approach may provide useful information for assessing flow redistribution under different pathological conditions, including potential applications in bypass surgical planning. To this end, we formulate a convex optimization-based strategy that combines flow conservation constraints with sparsity-prior regularization. This framework has been widely employed to address various networked flow problems, e.g., estimating traffic matrices from link loads in communication networks [23], detecting blockages in water distribution networks [24], and optimizing the traffic-power flow for the least-cost social optimum states [25]. Since blood transport within the cerebral vascular network also follows conservation principles and tends to minimize the total energy cost as characterized by Murray's law [26-28], the proposed approach is expected to effectively infer the source-sink flow matrix of the cerebral circulation governed by similar physical constraints.

Let $\mathbf{x} \in R^m$ denote the unknown vector of blood flow assigned to each of the $m$ candidate paths to terminal nodes which are first selected by identifying all possible paths using a depth-first-search (DFS) method, and then further refined by discarding those inconsistent with the quasi-steady blood flow directions predicted by the multiscale model. Based on the flow rate values $Q$ computed by the multiscale model throughout the vascular network, we then impose physiological flow conservations at each vessel segment ($Q_{line}$), the inflow sources nodes ($Q_{start}$) (e.g., LICA), and the outflow sink nodes ($Q_{end}$) (i.e., the terminal arteries) to construct the constraint system as follows:

$$\sum_{p \in \text{PathsThro}(l)} x_p = Q_{line}(l), \quad \forall l = 1, \ldots, L,$$

$$\sum_{p \in \text{PathsFrom}(i)} x_p = Q_{start}(i), \quad \forall i = 1, \ldots, S,$$

$$\sum_{p \in \text{PathsTo}(j)} x_p = Q_{end}(j), \quad \forall j = 1, \ldots, E, \tag{13}$$

where variable $x_p$ represents the actual volumetric blood flow through the $p$-th feasible path, along with $L$, $S$, and $E$ respectively denoting the numbers of segments, sources, and sinks. PathsThro($l$), PathsFrom($i$), and PathsTo($j$) respectively represent the sets of the feasible paths passing through the segment $l$, starting from node $i$, and terminating at node $j$. We concatenate the three constraint matrices into a unified system $W\mathbf{x} = \mathbf{b}$ by:

$$W = \begin{bmatrix} W_{line} \\ \lambda_1 W_{start} \\ \lambda_2 W_{end} \end{bmatrix}, \quad \mathbf{b} = \begin{bmatrix} \mathbf{Q}_{line} \\ \lambda_1 \mathbf{Q}_{start} \\ \lambda_2 \mathbf{Q}_{end} \end{bmatrix} \tag{14}$$

with

$$W_{line}(l, p) = \begin{cases} 1 & \text{if path } p \text{ passes line } l \text{ in forward flow direction,} \\ -1 & \text{if path } p \text{ passes line } l \text{ in reverse flow direction,} \\ 0 & \text{else,} \end{cases}$$

$$W_{start}(i, p) = \begin{cases} 1 & \text{if path } p \text{ starts from node } i, \\ 0 & \text{else,} \end{cases}$$

$$W_{end}(j, p) = \begin{cases} 1 & \text{if path } p \text{ ends at node } j, \\ 0 & \text{else.} \end{cases}$$

where $\lambda_1$ and $\lambda_2$ are weighting factors used to balance the influence of source and sink node constraints relative to segment-based constraints, thereby allowing flexibility in incorporating certain experimental data. For example, if reliable measurements of inflow rates at source nodes (e.g., ICA or VA from phase-contrast MRI) are available, a larger value of $\lambda_1$ can be used to enforce closer agreement with these observations.

However, the equation system that describe the conservation of flow across vascular segments is typically underdetermined due to the limited availability of internal measurements. In vascular trees, particularly within the cerebral circulation, blood flow distribution is normally assumed to follow principles consistent with Murray's law, which is derived by optimizing the total power expenditure required to pump blood through a single segment [26,29]. Motivated by this principle and the physiological preference for energetically efficient trunk-branch routes, we regularize the path-flow inversion by combining a sparsity prior with a pumping-cost penalty and solve it by the following manner:

$$\min_{\mathbf{x} \geq 0} \frac{1}{2} \| W\mathbf{x} - \mathbf{b} \|_2^2 + \delta \| \mathbf{x} \|_1 + \sigma \sum_{p=1}^{m} R_p x_p^2, \tag{15}$$

where $\delta > 0$ controls sparsity, i.e., minimizing the number of active paths, and $\sigma > 0$ weights the total costs with the path-resistance $R_p$ computed as the series sum of the resistance of all segments along path $p$. Thus, the inversion of path-flow is formulated as an $L_1$-regularized convex optimization problem with non-negativity constraints, which is solved in this study using the CVX modeling framework in MATLAB with its default interior-point solver (SDPT3 solver).

To further evaluate robustness of this optimization algorithm, we introduced multiplicative noise into the constraint vector $\mathbf{b}$: $\mathbf{b}' = \mathbf{b} \cdot (1 + \eta \xi)$, where $\xi \sim \text{Uniform}[-1,1]$ and $\eta \in \{0.05, 0.10\}$ represents a 5–10% perturbation of the flow results from the multiscale model. The perturbed $L_1$ problem (Eq. (15)) was solved 500 times, each with an independently perturbed input $\mathbf{b}'$, to evaluate the stability of the inferred path flows. The box plot and stability scores are reported in Supporting Information (S3 Text), which consistently show that the dominant flow-carrying paths exhibit narrow confidence bands and high stability scores (close to 1), indicating that the sparse-prior solutions are robust against perturbations. With the solution $\mathbf{x}$, we further define source-to-sink attributions by

$$\theta_{i,j} = \frac{\sum_{p \in \text{PathsFrom}(i) \cap \text{PathsTo}(j)} x_p}{\sum_{p \in \text{PathsTo}(j)} x_p}, \tag{16}$$

providing a quantitative descriptor of how inflow sources and vessel segments within cerebral structures distribute their contributions to distal vascular territories.

## 2.4. Computational workflow and parameters for models

The cerebral arterial network was adopted from works reported by [14], while the remaining extracerebral vasculature, including connecting segments such as aortic arch that link the heart to the cerebral circulation, was derived from the studies by Toro et al. [30]. The overall simulation process based on this cerebral vascular network is summarized in Fig 2.

Due to the extensive number of vessel segments, detailed geometrical parameters are not listed in this paper. Parameters for the heart model are from papers by Sun et al. [16] and by Liang et al. [17], and illustrated in Supporting Information (Table A in S2 Text). Additionally, the values of parameters related to the CAM refer to the studies conducted by Payne [31] and by [15], and are provided in Supporting Information (Table B in S2 Text). A summary of the network size and model parameters is provided in Supporting Information (S1 Table: Nomenclature table).

Given these parameters, a physiologically realistic simulation of cardiac–cerebral circulation can be implemented. The 1D domains are divided into grids with a length of $\Delta x = 0.1\,\text{cm}$, ensuring at least ten computational grids per vessel segment with the time step $\Delta t = 5 \times 10^{-4}$ s. The duration of a cardiac cycle $\tau$ is set to 1.0 s for iterative computations. Blood viscosity and density were respectively specified as $\mu = 0.0045$ Pa·s and $\rho = 1050\,\text{kg/m}^3$. The initial conditions were set as $Q(x, 0) = 0$ and $A(x, 0) = A_0$.

## 3. Numerical results

We used a CPU with 24 cores (Xeon Silver 4214R; Intel Corp.), with 64GB RAM, to implement the computation procedure including three categories of conditions: (1) baseline cerebral circulation within the complete CoW, (2) anatomical variations including the fetal-type PCAP1 and hypoplastic ACAA1 configurations to consider structural asymmetry and incomplete connectivity, and (3) pathological scenarios involving progressive stenosis applied to the LPCAP1 segment. For each case, the presented model iteratively converged to a quasi-steady state solution until the relative error $\varepsilon$ in Eq.(10) fell below 0.01, which serves as the stopping criterion for the iteration process. For validation, simulation results were compared with the clinical statistical results of the study by Zarrinkoob et al. [32] for 94 subjects.

### 3.1. Baseline cerebral flow distribution

Fig 3 provides a visualization of blood flow propagation over one cardiac cycle within the cerebral arterial network. The temporal evolution of flow highlights how pulsatile blood is distributed from the inflow arteries into the downstream territories. At early time points (t = 0.1 s), blood enters the cerebral vasculature through the VAs and ICAs. As time progresses, the flow gradually propagates toward distal branches, accompanied by a flow reduction in upstream vessels. This process reflects the spatiotemporal distribution of cerebral perfusion.g

For clarity, only the baseline case is depicted here, and videos illustrating the time-dependent flow propagation under all three conditions are available in the GitHub repository (see Data Availability Statement for details) to provide intuitive visualizations.

Subsequently, Fig 4 illustrates the baseline blood flow distribution within the entire cerebral vascular network and the complete CoW under normal physiological condition (i.e., no stenosis, with CAM considered).

In Fig 4a, terminal nodes are grouped according to their originating major cerebral arteries (RACA, LACA, RMCA, LMCA, RPCA, LPCA) and highlighted with distinct colors. A distinctive topological feature of this cerebral network is that the flows from LACA and RACA merge immediately after passing through their A2 segments, resulting in downstream terminal nodes being equally affected by both sides. Accordingly, outflow sink nodes originating from LACA and RACA are marked with the same color. For the complete CoW, it's worth noting that the communicating arteries (LPCoA, ACoA, RPCoA) carry small blood flow, especially ACoA shows negligible activation. This observation is consistent with experimental [9] and in-vivo [8] studies. Zhu et al. [9] reported that the pressure difference across both ends of PCoAs and ACoA in a complete CoW are almost zero, indicating that the anterior and posterior as well as the left and right circulations are relatively independent under normal physiological conditions.

In Fig 4b, simulated mean flow rates were compared with the clinical statistics reported by Zarrinkoob et al. [32] for the complete CoW. All simulation results (red filled circles) fall well within the clinical ranges defined by the mean±standard deviation (s.d.), confirming that the multiscale model can reproduce the physiological flow distribution across source vessels and downstream outlet vessels of the cerebral network in a statistically consistent manner.

PLOS Computational Biology

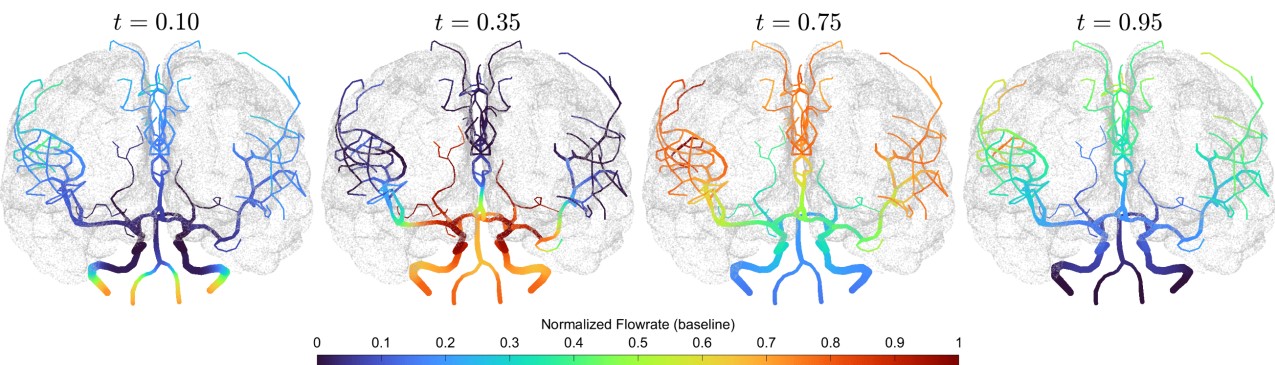

**Fig 3. Snapshots at representative time points illustrate the propagation of pulsatile flow over one cardiac cycle (one second) from inflow arteries to distal branches within the cerebral arterial network.** The line color at each spatial–temporal position $(x, t)$ represents the normalized values of blood flow $Q(x,t)/ \max_t |Q(x,t)|$.

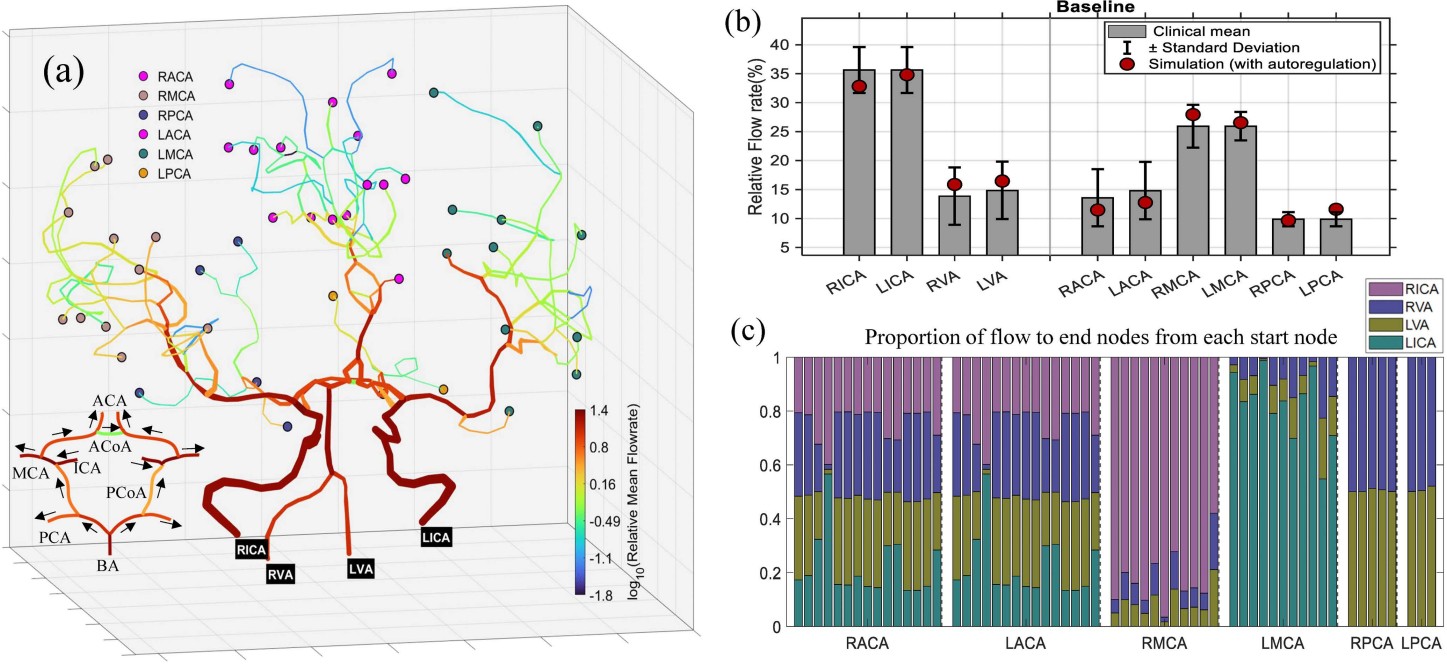

**Fig 4. (a) Spatial distribution of blood flow in the entire cerebral vascular network with the complete CoW.** The outflow sink nodes are grouped by their upstream segment originating from one of the six major arteries (RACA, LACA, RMCA, LMCA, RPCA, LPCA) and marked with different colors. Black arrows indicate the flow direction in the segments in the CoW. (b) Comparisons between simulated relative fractions of segments in CoW and clinical statistics reported by Zarrinkoob et al. [32]. (c) Normalized relative contributions of the four primary sources to each terminal nodes assigned to the six regions.

In Fig 4c, the normalized contribution from the four primary inflow sources (RICA, RVA, LVA, LICA) to each terminal node are quantified using Eq.(16). The results reveal heterogeneity in supply patterns. Specifically, the ACA territories (RACA and LACA) receive comparable contributions from both ICAs and VAs with minimal left–right asymmetry due to the flow merging from RACAA2 and LACAA2. The MCA territories (RMCA and LMCA) are predominantly perfused by their ipsilateral (i.e., same-side) ICAs, with only minor contributions from the VAs. In contrast, the PCA territories (RPCA and

LPCA) are almost entirely supplied by the VA inflows, exhibiting a symmetric distribution between the two sides. These observations suggest that the presented model not only reproduces the overall flow distribution but also provides detailed insights into the blood supply within the brain, reflecting how inflow sources and communicating branches of the CoW jointly shape the perfusion of distal vascular territories.

### 3.2. Anatomical variations with and without CAM

To investigate the hemodynamic consequences of the whole cerebral network with incomplete CoW and to assess the effects of CAM, the multiscale framework was employed with and without incorporating CAM for the CoW structures lacking PCAP1 (Fig 5) and ACAA1 (Fig 6).

Fig 5a illustrates the distribution of cerebral blood flow (CBF) changes from the baseline values presented in Fig 4. In this fetal-type PCAP1 configuration, ipsi-ICA exhibits the largest flow increase, while cont-ICA shows a minor rise, along with flow reductions in both LVA and RVA. Throughout this study, "ipsi-" and "cont-" respectively denote the ipsilateral and contralateral sides relative to the missing segment in the CoW. Importantly, all communicating arteries (PCoAs and ACoA) become activated, carrying markedly increased flows. In particular, both ipis-PCoA and ACoA even display flow reversals (denoted by the symbol ↻), where their flow direction becomes opposite to that under baseline conditions. These collateral adjustments effectively redistribute blood flow, thereby ensuring that most downstream terminal vessels maintain nearly unchanged perfusion despite the structural variation.

In Fig 5b, the simulated results obtained with and without considering CAM together with potential flow reversals, are compared against clinical measurements [32]. It is evident that the CAM-enabled simulations (red filled circles) align well within the clinical ranges (mean±s.d.), confirming the model's physiological consistency. In contrast, simulations without autoregulation (green triangles) exhibit substantial deviations in several vascular segments. The discrepancies are most pronounced in ipsi-PCAP2, which is connected to the missing PCAP1 segment, showing extreme ischemia, whereas cont-PCA presents severe hyperperfusion. Such unrealistic deviations arise from the absence of global pressure-balancing mechanisms within the entire cerebral vascular network, underscoring the essential roles of CAM and potential flow reversals in maintaining stable cerebral perfusion by physiologically consistent redistributions.

Moreover, as shown in Fig 5a, all four inflow sources undergo notable changes, illustrating that inflow sources and communicating branches of the CoW jointly determine the perfusion of distal vascular territories. Fig 5c further quantifies these adaptive effects: ACA territories exhibit only minor variations, while MCA and PCA territories undergo substantial reorganization. Specifically, the ipsi-MCA and PCA are almost exclusively supplied by the ipsi-ICA, whereas the cont-MCA and PCA proportions remain nearly unchanged. This highlights the compensatory function of the CoW: the full activation of PCoA and its associated flow reversal modulate perfusion toward MCA and PCA regions, whereas the activation of ACoA and its redirected flow ensure perfusion symmetry between the hemispheres.

Fig 6 provides the simulation results for the case with a missing or hypoplastic A1 segment of ACA. In contrast to the fetal-type PCAP1 case shown in Fig 5, where ipsi-ICA exhibited a pronounced flow increase, the present configuration shows moderate decreases in flow rates across ipsi-ICA and both VAs, while cont-ICA displays a substantial compensatory increase. From the enlarged CoW view, ipsi-ICA only supplies the MCA and PCA territories. Consequently, the total inflow and the flow through the communicating arteries on the ipsilateral side are adaptively decreased to prevent hyperperfusion in these regions. As illustrated in Fig 6a, the ipsi-MCA and PCA, together with their downstream vessels, maintain stable perfusion, indicating that autoregulatory adjustments within the inflow pathways effectively stabilize cerebral blood flow in the ipsilateral hemisphere. This behavior is also supported by Fig 6b. When CAM is neglected, the simulated flow rate of ipsi-MCA (green triangles) greatly exceeds the clinical range, whereas the CAM-enabled results (red filled circles) align well with clinical observations, confirming that cerebral autoregulation is essential for maintaining physiologically consistent perfusion under different CoW structures.

**Fig 5. (a) Cerebral blood flow (CBF) changes from the baseline values of the entire cerebral vascular network with a fetal-type PCAP1 in CoW structure, where the PCAP1 segment is disconnected from BA and the PCAP2 segment originates from the ipsi-ICA through ipsi-PCoA** [33]. The outflow (terminal) nodes are grouped by their upstream major arteries (RACA, LACA, RMCA, LMCA, RPCA, LPCA), and marked with distinct colors. The symbol ↻ indicates that the flow direction in the segment is opposite to that under baseline conditions. (b) Comparisons between simulated relative fractions of the CoW segments and the clinical statistics (mean ± s.d.) reported by Zarrinkoob et al. [32]. Red filled circles and green triangles respectively represent simulations with and without considering the CAM. (c) Normalized relative contributions of the four primary inflow sources to each terminal nodes assigned to the six major regions. Ipsi: ipsilateral; cont: contralateral.

In addition, Fig 6a reveals that the flow rate in the pathways supplying the ACA territory from cont-ICA becomes noticeably increased, which is quantitatively confirmed in Fig 6c indicating that the ACA region becomes almost entirely supplied by the cont-ICA with minor contributions from both VAs. This redistribution fully activates ACoA, which exhibits a great flow increase and flow reversal (denoted by ↻), thereby ensuring balanced perfusion between the two hemispheres. Moreover, PCoAs play a relatively limited role in this configuration, and ipsi-PCoA even exhibits a slight decrease in flow from BA toward ipsi-MCA, likely reflects an autoregulatory adjustment acting to prevent excessive perfusion in the ipsilateral hemisphere.

In Fig 6b, when CAM is neglected, the simulated cont-ACA (green triangles) exhibits severe ischemia due to the unregulated low inflow across cont-ICA, whereas incorporating CAM restores physiologically reasonable flow rates for both cont-ICA and cont-ACA (red filled circles), in good agreement with the clinical statistics provided by Zarrinkoob et al. [32]. Furthermore, the computed flow through the hypoplasia ACAA1 segment is close to zero due to its small diameter (defined as less than 0.8 mm), and the corresponding measured value is also zero. Therefore, the bar is not visible, and only the markers (circle and triangle) near zero are displayed to preserve axis alignment with the bar plots in Figs 4b and 5b.

In addition to the separate analyses of each anatomical configuration presented before, Fig 7 further compares the hemodynamic differences within the CoW under the two variation scenarios relative to the baseline condition. As shown in Fig 7d, the simulated CBF variations (red for increase and blue for decrease) agree well with the clinical data from Zarrinkoob et al. [32], with all simulated data points (circles) falling within the physiological range. Although minor sign discrepancies are observed in a few vessels (e.g., at cont-MCA), their values remain consistent with the measured uncertainty. confirming that the model accurately captures the redistribution of cerebral blood flow.

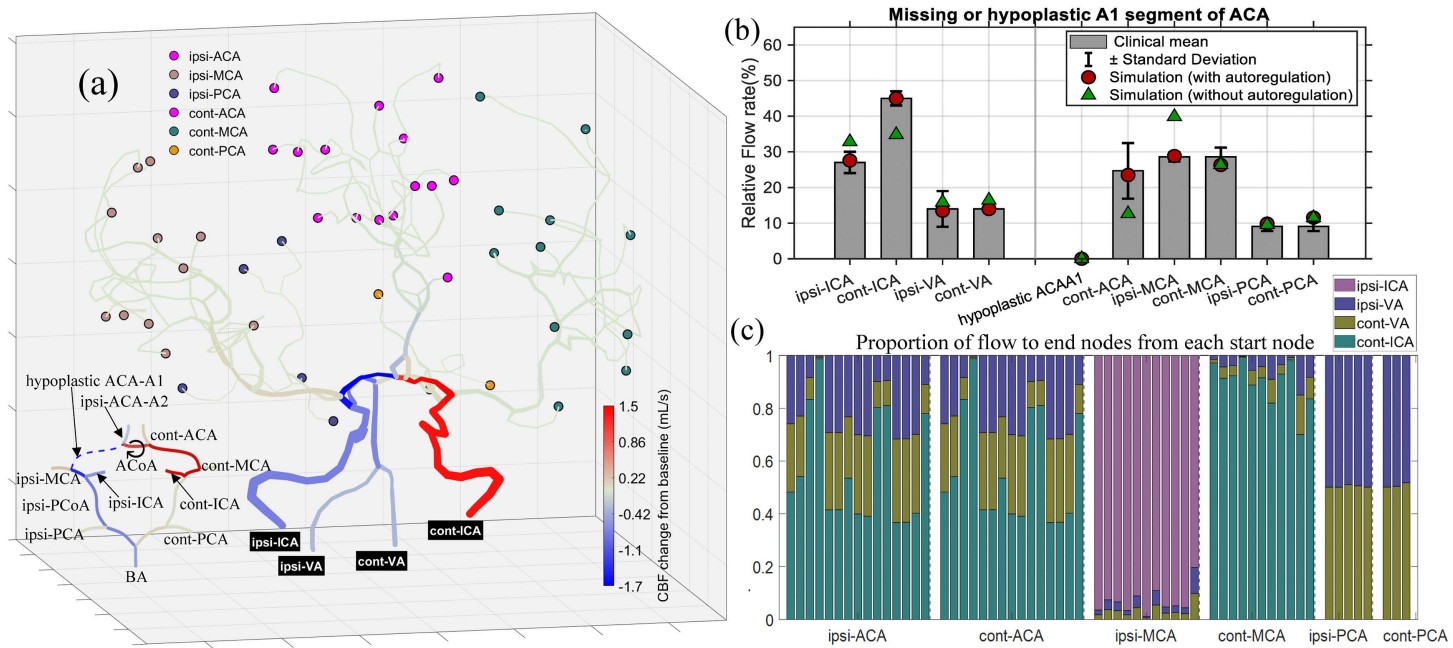

**Fig 6. (a) Cerebral blood flow (CBF) changes from baseline condition for the cerebral vascular network with a missing or hypoplastic A1 segment of ACA in CoW structure, where hypoplasia is defined as a vessel diameter smaller than 0.8 mm [34].** Outflow (terminal) nodes are grouped according to their upstream major arteries (RACA, LACA, RMCA, LMCA, RPCA, LPCA), and displayed with distinct colors. The symbol ↻ indicates that the flow direction in the segment is opposite to that under baseline conditions. (b) Comparisons between simulated relative fractions and clinical statistics (mean±s.d.) by Zarrinkoob et al. [32]. Red filled circles and green triangles respectively represent simulations with and without considering the CAM. (c) Normalized relative contributions of the four primary inflow sources to each terminal nodes assigned to the six major regions. Ipsi: ipsilateral; cont: contralateral.

Given this good agreement in predicting the flow rates of the non-communicating segments in the CoW, the systemically computed results for the communicating arteries that lack direct measurements can therefore be regarded as reliable. Fig 7b presents the baseline flow rate (mL/s) together with the relative CBF variations (%) for the two incomplete CoW structures, where the labels with "**" in the bar plots (e.g., ipsi-PCoA**) correspond to vessels exhibiting a reversal in flow direction and being marked by the symbol ↻ in the CoW schematics (Fig 7a and 7c). The results reveal that ACoA is the most sensitive collateral pathway, exhibiting significant flow amplification in both configurations, while PCoAs display structure-dependent responses, being strongly activated under the fetal-type PCAP1 configuration but showing reduced participation when ACAA1 segment is missing. These findings are consistent with the clinical observations provided by Zhu et al. [9], which demonstrated that ACoA is the most sensitive index to ipsi-ICA morphological changes and serves as the primary anterior compensatory channel, whereas PCoAs are important for the anterior-posterior collateral coupling.

Overall, these analyses demonstrate that the presented multiscale model incorporating CAM effectively captures the intrinsic capability of the CoW to maintain stable cerebral perfusion through adaptive collateral regulation under different anatomical variation conditions. To further bridge the gap between the physiological baseline and the extreme anatomical variations that are topologically equivalent to complete occlusions, the following section extends this analysis to investigate the continues transition of hemodynamic redistribution under progressive stenosis.

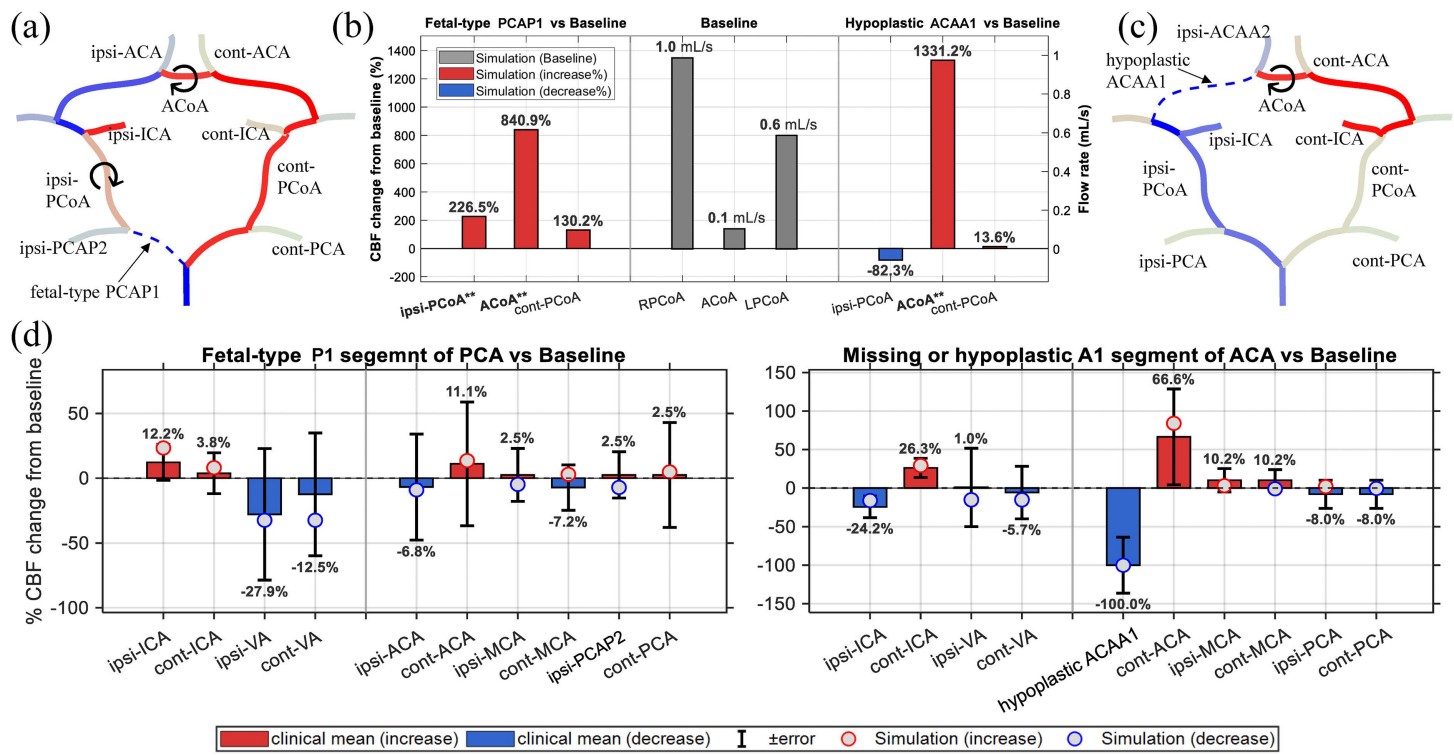

**Fig 7. Comparison of the CoW hemodynamic responses under two anatomical variation scenarios relative to the baseline condition. (a, c)** Schematic visualization of CBF redistribution for the CoW configurations with fetal-type PCAP1 and hypoplastic ACAA1, where the symbol ↻ indicates that the flow direction in the segment is opposite to that under baseline conditions and the color scale follows Figs 5-6. **(b)** Predicted baseline flow rates (mL/s) in the communicating arteries (PCoAs, ACoA) and their relative CBF changes (%) from the baseline for the two incomplete CoW structures. Bars labeled with "**" denote the same vessels marked by ↻ in **(a, c)**, indicating flow direction reversal. **(d)** Comparison of simulated CBF changes (circles) with clinical means±s.d. from Zarrinkoob et al. [32]. Red and blue respectively indicate increase and decrease.

## 4. Application and discussion

### 4.1. Flow redistribution under stenosis progression

In this subsection, Fig 8 presents the simulated redistribution of CBF changes within the CoW as the degree of stenosis $R_s$ (defined by Eq. (12)) in the RPCAP1 segment progressively increases. The simulation results were obtained by the multiscale model coupled with the localized stenosis formulation through Eq.(11). By gradually increasing $R_s$, the model allows quantitative validation of the hemodynamic redistribution process, verifying that the predicted collateral pathways, activation sequence, and flow patterns converge to those observed in anatomical variations. It ensures the physical continuity between normal, pathological narrowing (stenosis), and structurally incomplete CoW states within a unified multiscale framework.

As the stenosis increases from 20% to 80%, the overall flow in the ipsilateral circulation gradually decreases, while the contralateral side compensates with enhanced inflows. The hemodynamic asymmetry between hemispheres is progressively balanced through the communicating arteries, reflecting their increasing activations. Table in Fig 8 quantitatively presents the progressive transition from the baseline to the fetal-type PCAP1 configuration as stenosis severity increases, illustrating the continuous adaptation of collateral flows.

As discussed previously for Fig 4, the collateral mechanism of the communicating vessels in the CoW are almost inactive under the normal conditions [9], showing small flow and near-zero pressure differences across their ends. Compared

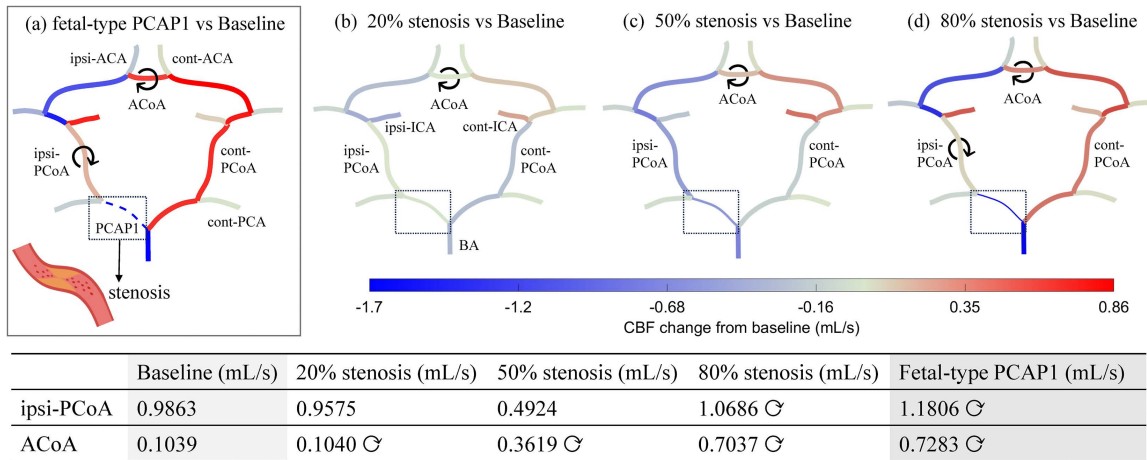

| | Baseline (mL/s) | 20% stenosis (mL/s) | 50% stenosis (mL/s) | 80% stenosis (mL/s) | Fetal-type PCAP1 (mL/s) |
|---|---|---|---|---|---|
| ipsi-PCoA | 0.9863 | 0.9575 | 0.4924 | 1.0686 ↻ | 1.1806 ↻ |
| ACoA | 0.1039 | 0.1040 ↻ | 0.3619 ↻ | 0.7037 ↻ | 0.7283 ↻ |
| cont-PCoA | 0.5870 | 0.3188 | 0.4010 | 1.2198 | 1.2785 |

\* The symbol ↻ indicates that the flow direction is opposite to that under the baseline condition.

**Fig 8. Flow redistribution under stenosis conditions. (a–d)** Simulated CBF changes from baseline for the fetal-type PCAP1 case and 20%, 50%, and 80% stenosis conditions applied to the ipsilateral feeding artery. Blue and red indicate flow decrease and increase, respectively. The dashed lines surrounded by dotted box denote the segment with stenosis. The table lists baseline flow rates (mL/s) of the major communicating arteries and their corresponding values under different stenosis levels. The symbol ↻ indicates that the flow direction in the segment is opposite to that under baseline condition.

with this state, as shown in Fig 8b, a mild stenosis of 20% produces slight redistribution of flow, mainly affected by the inflow adjustments (ICAs and BA), implying adequate upstream compensation regulation. Both the ACoA and PCoAs display only minor changes in their cross-flow, reflecting the unperturbed hemodynamic balance of the CoW with mild stenosis. Nevertheless, as listed in the accompanying table, although the flow rate through ACoA remains nearly unchanged value, its direction reverses, indicating an early activation. This finding also agrees with the experimental conclusion that the ACoA is the most sensitive index of unilateral morphological alterations and immediately carries flow from the healthy side to the affected side in response to stenosis [9].

When the stenosis degree increases to 50% (Fig 8c), the ACoA begins to exhibit moderate pressure-driven reverse flow, whereas partial activation of the ipsi-PCoA appears, showing the onset of cross-hemispheric redistribution. At 80% stenosis (Fig 8d), both ACoA and PCoAs are fully activated. Flow reversal (signed by ↻ in Fig 8) emerges in the ACoA and ipsi-PCoA, ensuring sufficient collateral perfusion to the affected hemisphere. In contrast, cont-PCoA shows an increase in flow magnitude while maintaining its original direction, reflecting a secondary activation state that supports, rather than redirects, collateral flow. These findings related to PCoAs are consistent with experiments by Zhu et al. [9] and with clinical transcranial Doppler (TCD) measurements by Hoksbergen et al. [8], both of which demonstrate that when unilateral stenosis occurred, ipsi-PCoA becomes an essential collateral pathway supplying the anterior circulation and is fully activated under moderate to severe stenosis (exceeds approximately 40%), whereas cont-PCoA contributes limited support even under complete occlusion.

These consistencies confirm that the presented model accurately reproduces the physiological collateral availability of the CoW, with early ACoA activation followed by ipsi-PCoA compensation, and thus provides a mechanistic explanation bridging experimental observations and in-vivo hemodynamics.

### 4.2. Source-to-sink flow contribution mapping and perfusion sensitivity under CoW variations

Figs 4c and 5c illustrated the normalized relative contributions of the four primary inflow arteries (RICA, LICA, RVA, LVA) to each terminal node based on the path–flow inversion framework described in Section 2.3. Here, Fig 9 further visualizes

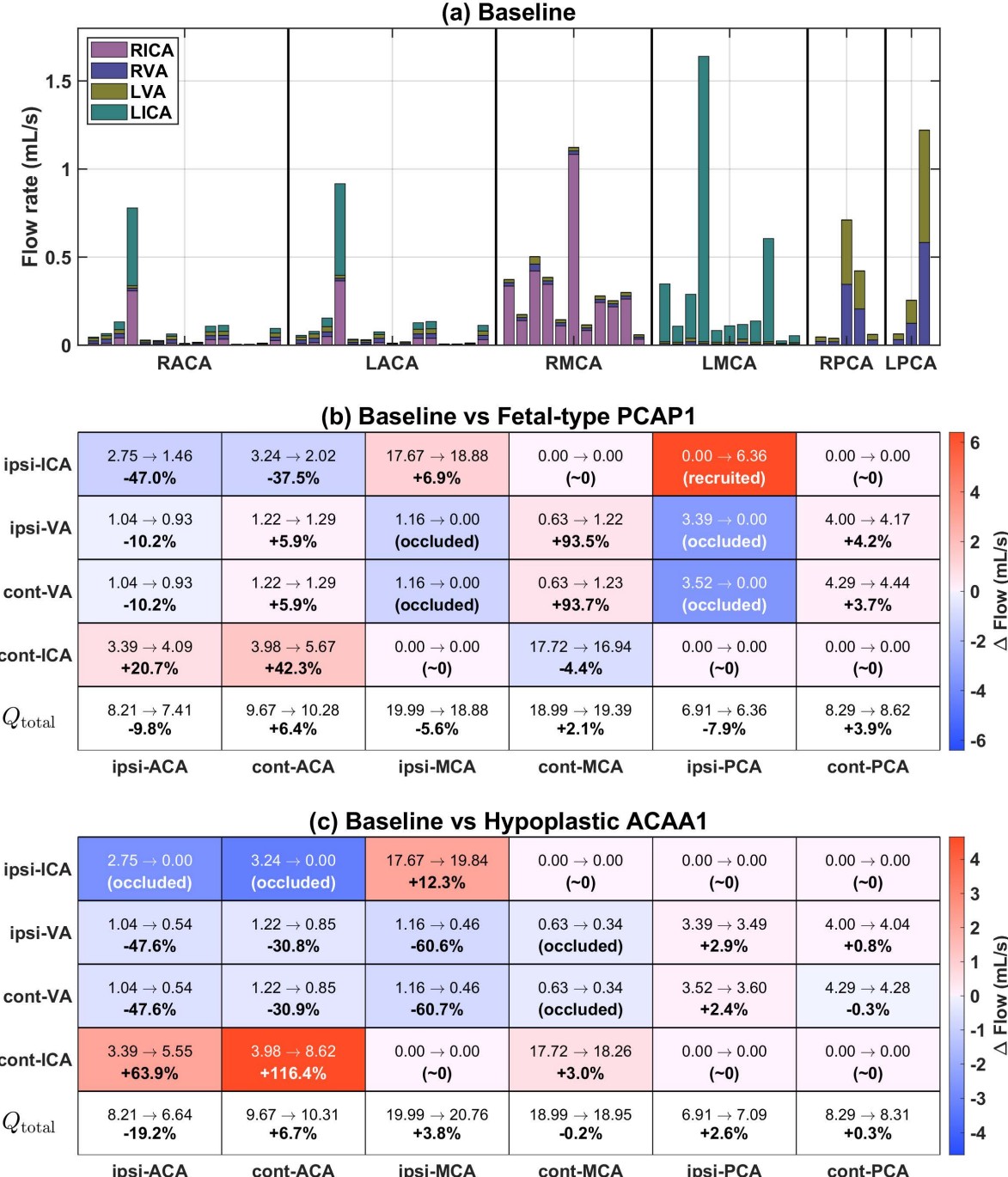

**Fig 9. Source-to-sink flow contribution mapping across different CoW configurations. (a)** Baseline: flow contribution (mL/s) of inflow arteries (RICA, LICA, RVA, LVA) to each terminal node grouped into six regions (RACA, LACA, RMCA, LMAC, RPCA, LPCA). **(b)-(c):** relative flow changes from baseline under (b) fetal-type PCAP1 and **(c)** Hypoplastic ACAA1 configurations. Arrows in each entry represent the contribution changes (mL/s and %) from baseline to different anatomical variations. Entries labeled "occluded" indicate that the corresponding pathway becomes inactive under the altered CoW configuration, whereas "recruited" denotes a newly activated pathway that is absent under baseline condition, reflecting the deactivation or recruitment of specific collateral supply pathways connecting inflow arteries to downstream territories.

the absolute source-to-sink flow contributions (mL/s) at baseline and the normalized relative changes from the baseline (%) under the two occlusion conditions, thereby providing a comprehensive depiction of how each terminal node responds to structural variations within the CoW.

Under the baseline conditions (Fig 9a), the regional flow distribution exhibits a clear lateral dominance, with the ACAs and MCAs primarily perfused by the ipsilateral ICAs. The VAs mainly supply the posterior circulation (PCAs) through the BA. These findings are consistent with the well-recognized distinction between the anterior and posterior circulations of the brain [35,36], which are primarily served by the carotid system (ICAs and their branches) and vertebrobasilar system (VAs and BA), respectively. In addition, ACA and PCA regions receive nearly symmetric inflows from both the right and left circulations, highlighting the well-balanced bilateral perfusion with negligible cross-hemispheric compensation under normal physiological conditions.

For the two CoW configurations shown in Figs 9b-c, the redistribution of source contributions reveals how collateral pathways are recruited to maintain cerebral perfusion. When the CoW configurations is altered from the complete CoW (baseline), the ICAs exhibit opposite trends of contribution to the ACA territory. The flow from the ipsi-ICA consistently decreases, whereas that from the cont-ICA increases, reflecting a compensatory redistribution mediated by the ACoA. In contrast, contributions from VAs remain symmetric between the left and right hemispheres, suggesting that the bilateral balance is primarily achieved through the regulation of ICAs and the activation of ACoA rather than by the posterior circulation.

Particularly, in the fetal-type PCAP1 configuration (Fig 9b), the ICAs and VAs show almost opposite and complementary patterns of redistribution across the MCAs and PCAs, implying a strong coupling between the anterior and posterior circulations. On the ipsilateral side, the ICA increases its flow supply to both the ipsilateral MCA and PCA regions, while the contribution of VAs decreases, indicating recruitment of collateral pathways through the ipsilateral PCoA. Conversely, on the contralateral side, the VAs contribution increases as the flow from the contralateral ICA decreases in the MCA territory or remains nearly unchanged in the PCA territory, also reflecting a dynamic posterior-to-anterior compensatory adjustment mediated by the communicating arteries.

In contrast, such anterior-posterior coupling is absent in the hypoplastic ACAA1 configuration (Fig 9c). In this case, the ACoA becomes fully activated and serves as the dominant compensatory pathway, while the PCoAs play a relatively limited role. As a result, perfusion to the ipsi-MCA territories is primarily maintained by the ipsi-ICA, whereas the PCA territories remain predominantly supplied by the vertebrobasilar system with relatively symmetric contributions from both VAs. Consequently, the redistribution of inflow contributions occurs mainly within the anterior circulation with limited interaction between the anterior and posterior pathways.

Importantly, the appearance or disappearance of specific source-to-sink contributions in Fig 9 reflects the recruitment or suppression of collateral supply pathways revealed by the path–flow inversion framework. This decomposition therefore provides an interpretable representation of how structural variations in the CoW translate into functional redistribution of cerebral perfusion across territories. The heterogeneous sensitivities of terminal territories to different inflow sources highlight the adaptive perfusion capacity of the network to anatomical variations, and may thus provide a quantitative basis for future diagnostic assessment and therapeutic planning in cerebrovascular diseases.

### 4.3. Role and clinical relevance of cerebral autoregulation

The incorporation of CAM is a critical component of the present framework and is essential for achieving physiologically plausible cerebral perfusion under both anatomical variations and pathological stenosis. As demonstrated in this study, simulations that neglected autoregulatory adjustment resulted in marked deviations from clinically observed flow distributions, including unrealistic hypo- or hyperperfusion in specific vascular territories. In contrast, the CAM-enabled simulations consistently reproduced flow patterns that aligned with experimental measurements and in vivo clinical data, underscoring the necessity of autoregulatory control for realistic cerebrovascular modeling.

From a clinical standpoint, cerebral autoregulation represents the brain's intrinsic capacity to stabilize regional blood flow in the face of fluctuations in perfusion pressure and upstream resistance. The present results suggest that the effectiveness of collateral pathways through the CoW cannot be fully interpreted without accounting for concurrent autoregulatory modulation of peripheral vascular resistance. This finding is consistent with clinical observations that patients with similar degrees of arterial stenosis may exhibit markedly different perfusion states and ischemic tolerance, depending on the integrity of their autoregulatory reserve.

It should be noted that the CAM parameters adopted in this study represent a generalized autoregulatory response derived from previously published models. In clinical populations, however, autoregulatory capacity is known to be heterogeneous and may be impaired in conditions such as advanced age, diabetes mellitus, chronic hypertension, or prolonged cerebral ischemia. While the present formulation captures the fundamental role of autoregulation in shaping collateral flow redistribution, future extensions incorporating patient-specific or condition-dependent autoregulatory characteristics may further enhance the clinical interpretability of the model and improve its applicability to individualized cerebrovascular risk assessment and treatment planning.

### 4.4. Clinical implications and decision-making relevance

From a clinical perspective, the present framework provides a functional interpretation of collateral circulation that extends beyond anatomical assessment alone. By capturing the sequential recruitment of collateral pathways—such as early flow reversal in the anterior communicating artery followed by engagement of the ipsilateral posterior communicating artery under increasing stenosis—the model identifies hemodynamic transition states that may reflect progressive stress on cerebral perfusion rather than luminal narrowing severity per se.

Importantly, the results indicate that effective cerebral perfusion emerges from the dynamic interplay between proximal collateral pathways, autoregulatory modulation of peripheral resistance, and redistribution of inflow contributions across vascular territories. This finding is consistent with clinical observations that patients with comparable degrees of arterial stenosis may exhibit markedly different ischemic tolerance and outcomes. The source-to-sink flow analysis further offers a quantitative means to interpret which vascular territories are functionally vulnerable and which collateral routes are already maximally recruited under pathological conditions. For example, when evaluating potential bypass strategies for reduced perfusion observed in a specified downstream territory, the path-flow inversion can identify which pathways play a critical role in maintaining perfusion and how surgical modification of the network may redistribute source contributions across territories, thereby helping to restore a physiologically balanced perfusion pattern consistent with the baseline flow distribution. Although explicit clinical thresholds are not proposed in the present study, the framework provides a quantitative platform for future patient-specific investigations aimed at functional stratification of cerebrovascular disease. Such applications may support risk assessment and treatment planning for revascularization strategies by characterizing residual collateral capacity and identifying territories at risk under further hemodynamic compromise.

### 5. Conclusion

In this work, we proposed and validated a 0D-1D multiscale model coupled with a CAM to physiologically characterize the cardio-cerebrovascular circulation. The developed framework integrates systemic hemodynamics, medical-image-based cerebral vascular reconstruction, and cerebral autoregulation functionality to assess the collateral capacity of the CoW under anatomical variations and progressive stenosis. This approach provides valuable insights into how collateral pathways dynamically adapt to topological and resistive changes within the vascular network.

The results demonstrate that the communicating arteries of the CoW remain nearly inactive under the normal physiological conditions, whereas anatomical variations and progressive stenosis sequentially activate collateral pathways. The ACoA is activated first, initiating cross-hemispheric flow redistribution through flow reversal, followed by the activation of

the ipsilateral PCoA that supports posterior-to-anterior compensation at more severe stenosis stages. These hemodynamic patterns are consistent with experimental and in-vivo observations.

Furthermore, the source-to-sink analysis reveals that perfusion adaptability follows a region-dependent hierarchical pattern, where the anterior circulation driven by the ICAs dominates the primary compensation, while the posterior circulation supplied by the VAs provides secondary support when the anterior pathways become insufficient for the MCA and PCA regions.

Overall, the proposed framework introduces several innovations. It systematically integrates the 0D heart model as the inlet boundary and the CAMs as the outlet boundary within a unified resistance-matrix formulation of the cerebral vascular network. This integration enables consistent simulations of cerebral circulation without requiring any locally measured or prescribed cerebral flow or pressure data. As a result, the model can dynamically and autonomously capture blood flow redistribution arising from topological and morphological variations in the cerebral vasculature, including arterial occlusion and progressive stenosis scenarios. In addition, this study presents a convex-optimization-based flow inversion strategy, which quantifies the flow contributions from the four major cerebral inflow arteries (VAs and ICAs) to downstream territories and evaluates their sensitivity to anatomical changes in the cerebral vascular network.

However, several limitations of the present study should be acknowledged. In the present framework, the 0D heart model is connected to the cerebral arterial network through lumped resistance elements (Fig 1c), and thus pulse wave propagation along the aorta is not explicitly resolved, whereas Liang et al. [17] incorporated a 1D representation of the aorta. Consequently, this study may underestimate detailed waveform distortions during transmission through the aorta, although it still provides adequate inflow conditions for investigating cerebral flow redistribution and collateral circulation. In terms of modeling, the parameters used in the CAMs were adopted from [15], assigning identical values to the terminal nodes within the same territory (MCA, ACA, or PCA). This simplification may not fully capture the local, physiological heterogeneity of autoregulatory behaviors across different terminal vessels reconstructed from medical images. In addition, parameters outside the CAM model may also influence cerebral flow dynamics, e.g., vascular stiffness plays an important role in pulse wave propagation and pressure transmission along the arterial tree. In the present model, uniform stiffness parameters (Young's modulus $E_s$ and wall thickness $h$) are assumed for arteries, whereas in reality arterial stiffness may vary with age, disease state, and anatomical location, which may influence flow redistribution patterns. Moreover, to the best of our knowledge, this study represents the first attempt to introduce the convex-optimization-based flow inversion algorithm to the blood-flow vascular network. Although this approach follows the energy-minimization principle underlying Murray's law and has been widely applied in other fields for networked systems such as communication networks [23], it has not yet been quantitatively validated with experimental or clinical measurements for cerebral hemodynamics. Consequently, further validation focusing on local hemodynamic characteristics within specific cerebral vessels is still required.

In addition, the present framework primarily focuses on proximal collateral pathways mediated by the CoW. Distal leptomeningeal anastomoses between peripheral cortical arteries, which are known to play a critical role in maintaining cerebral perfusion in advanced ischemic states when the CoW collaterals become insufficient, are not explicitly represented as anatomical connections in the current vascular network.

Importantly, however, the proposed model already incorporates several key physiological mechanisms that provide a functional basis for resolving such distal collateral behavior. The inclusion of cerebral autoregulation enables dynamic reduction of peripheral vascular resistance, the source-to-sink flow analysis quantitatively characterizes shifts in inflow contributions, and the territory-level perfusion framework ensures stabilization of regional cerebral blood flow under pathological conditions. With the explicit incorporation of inter-territorial peripheral arterial connections representing leptomeningeal anastomoses, the present multiscale framework could be naturally extended to provide a more comprehensive and physiologically faithful assessment of cerebral collateral capacity, potentially yielding clinically valuable insights into ischemic vulnerability and treatment planning.

In future work, this framework will be extended to patient-specific simulations that combine anatomical variations with pathological occlusion or stenosis, enabling quantitative assessment of their coupled effects on cerebral flow redistribution, activation sequences of the communicating arteries, and collateral pathways. Such extensions, including the incorporation of distal collateral pathways, are expected to provide clinically meaningful information for optimizing surgical and revascularization strategies for cerebrovascular diseases.

## Supporting information

**S1 Text. Pseudocode of the proposed CAM-incorporated path-flow inversion framework.**
(PDF)

**S2 Text. Parameter values for the heart model and CAM.**
(PDF)

**S3 Text. Robustness testing of the proposed sparse-prior inversion algorithm.**
(PDF)

**S1 Table. Nomenclature and list of symbols.**
(PDF)

## Author contributions

**Conceptualization:** Jiawei Liu, Hiroshi Suito.

**Formal analysis:** Jiawei Liu.

**Funding acquisition:** Hiroshi Suito.

**Investigation:** Atsushi Kanoke, Hidenori Endo, Kuniyasu Niizuma.

**Methodology:** Jiawei Liu.

**Project administration:** Hiroshi Suito.

**Resources:** Atsushi Kanoke, Hidenori Endo, Kuniyasu Niizuma.

**Software:** Jiawei Liu.

**Supervision:** Hiroshi Suito.

**Validation:** Jiawei Liu, Atsushi Kanoke, Hidenori Endo, Kuniyasu Niizuma.

**Writing – original draft:** Jiawei Liu, Kuniyasu Niizuma, Hiroshi Suito.

**Writing – review & editing:** Jiawei Liu, Atsushi Kanoke, Hidenori Endo, Kuniyasu Niizuma, Hiroshi Suito.

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
