## [Decision Letter · Decision Letter 0]

28 Jan 2026

PCOMPBIOL-D-25-02693

Multiscale modeling of blood circulation with cerebral autoregulation and network pathway analysis for hemodynamic redistribution in the vascular network with anatomical variations and stenosis conditions

PLOS Computational Biology

Dear Dr. Liu,

Thank you for submitting your manuscript to PLOS Computational Biology. After careful consideration, we feel that it has merit but does not fully meet PLOS Computational Biology's publication criteria as it currently stands. Therefore, we invite you to submit a revised version of the manuscript that addresses the points raised during the review process.

We look forward to receiving your revised manuscript.

Kind regards,

Hamidreza Mortazavy Beni

Academic Editor

PLOS Computational Biology

Stacey Finley

Section Editor

PLOS Computational Biology

**Journal Requirements:**

At this stage, the following Authors/Authors require contributions: LIU JIAWEI. Please ensure that the full contributions of each author are acknowledged in the "Add/Edit/Remove Authors" section of our submission form.

4) We notice that your supplementary Figures, and information are included in the manuscript file. Please remove them and upload them with the file type 'Supporting Information'. Please ensure that each Supporting Information file has a legend listed in the manuscript after the references list.

**Reviewers' comments:**

Reviewer's Responses to Questions

**Comments to the Authors:**

Reviewer #1: This is an excellent manuscript that is well written and presented with clarity. The figures are superb and highlight the most important findings. The methods are described with unusual detail and the manuscript advances the field in a useful way. Strengths and weaknesses of the approach are discussed.

The only slight concern is that the authors do not appear to have made their source code freely available. The data are linked through URLs but this reviewer did not find information about the computational code itself.

Reviewer #2: The review is uploaded as an attachment.

Reviewer #3: See attached.

**Have the authors made all data and (if applicable) computational code underlying the findings in their manuscript fully available?**

Reviewer #1: **No:**Data are provided but the source code does not appear to be publicly available.

Reviewer #2: **No:**I have some difficulty going through the file artery.csv since there is only one column. If an explanation could be added to this, it would be nice.

Reviewer #3: **No:**Links to data and other resources were not working for this reviewer.

PLOS authors have the option to publish the peer review history of their article (what does this mean?). If published, this will include your full peer review and any attached files.

Reviewer #1: No

Reviewer #2: No

Reviewer #3: No

**Figure resubmission:**
---

## [Decision Letter · Decision Letter 1]

4 May 2026

Dear Dr. Liu,

We are pleased to inform you that your manuscript 'Multiscale modeling of blood circulation with cerebral autoregulation and network pathway analysis for hemodynamic redistribution in the vascular network with anatomical variations and stenosis conditions' has been provisionally accepted for publication in PLOS Computational Biology.

Best regards,

Hamidreza Mortazavy Beni

Academic Editor

PLOS Computational Biology

Stacey Finley

Section Editor

PLOS Computational Biology

Reviewer's Responses to Questions

**Comments to the Authors:**

Reviewer #1: No further comments

Reviewer #2: The authors have addressed all my concerns. I have no further questions.

Reviewer #3: I am satisfied with the revisions. Thank you for making the manuscript more accessible through your edits.

**Have the authors made all data and (if applicable) computational code underlying the findings in their manuscript fully available?**

Reviewer #1: Yes

Reviewer #2: **No:**

Reviewer #3: Yes

PLOS authors have the option to publish the peer review history of their article (what does this mean?). If published, this will include your full peer review and any attached files.

Reviewer #1: No

Reviewer #2: No

Reviewer #3: No

---

## [Editor Report · Acceptance letter]

PCOMPBIOL-D-25-02693R1

Multiscale modeling of blood circulation with cerebral autoregulation and network pathway analysis for hemodynamic redistribution in the vascular network with anatomical variations and stenosis conditions

Dear Dr Liu,

I am pleased to inform you that your manuscript has been formally accepted for publication in PLOS Computational Biology. Your manuscript is now with our production department and you will be notified of the publication date in due course.

With kind regards,

Anita Estes
